# Investigation of Wind Field Parameters for Long-Span Suspension Bridge Considering Deck Disturbance Effect

**DOI:** 10.3390/s25216503

**Published:** 2025-10-22

**Authors:** Yonghui Zuo, Xiaoyu Bai, Rujin Ma, Zichao Pan, Huaneng Dong

**Affiliations:** 1School of Transportation, Southeast University, Nanjing 210096, China; 2CCCC Highway Bridges National Engineering Research Centre Co., Ltd., Beijing 100120, China; 3College of Civil Engineering, Tongji University, Shanghai 200092, China; 4Jiangsu Expressway Engineering Maintenance Co., Ltd., Nanjing 211106, China

**Keywords:** wind field parameters, long-span bridge, deck disturbance effect, data cleaning

## Abstract

This study investigates the wind field characteristics of long-span suspension bridges, with a particular focus on the disturbance effects introduced by the bridge deck on wind measurements. Field data are collected using anemometers installed on both the upstream and downstream sides at the midspan of the bridge girder. A comparative analysis of these measurements reveals notable discrepancies attributable to deck-induced flow disturbances. To systematically assess these effects, the disturbed wind parameters are identified, and their spatial distribution patterns are examined. A statistical model is then developed to quantify and correct the disturbance influence. This model isolates the disturbance component and establishes empirical correlations between the disturbed and actual wind parameters. The results confirm that the proposed correction approach effectively reduces measurement bias caused by deck interference, thereby enabling more accurate wind load evaluation for long-span suspension bridge structures.

## 1. Introduction

Long-span bridges, due to their inherently low stiffness and damping characteristics, are particularly vulnerable to wind-induced vibration, e.g., flutter, buffeting, and vortex-induced vibration [1,2,3]. Ensuring aerodynamic stability and structural safety under complex wind environments is therefore a critical aspect of the design and maintenance of these bridges. A comprehensive understanding of site-specific wind field characteristics—such as mean wind speed, turbulence intensity, and wind directionality—is essential to support reliable wind-resistant designs [4,5].

With the advancement of sensing and data acquisition technologies, long-span bridges are increasingly equipped with structural health monitoring (SHM) systems that continuously capture wind and structural response data in real time [6]. These monitoring systems have enabled extensive investigations into the wind environment, allowing for statistical characterization [7,8,9] of in situ wind fields and providing valuable insights into the wind load characteristics of long-span bridges [10,11].

Building upon these foundational capabilities, the extensive application of SHM wind data has yielded significant insights into the complex wind environments surrounding long-span bridges. Numerous investigations have leveraged these monitoring systems to characterize site-specific wind patterns, extreme wind events, and their implications for structural performance [12,13]. Studies have utilized SHM data to establish probabilistic wind models for bridge design [7,9], investigate wind–structure interaction phenomena [5,8], and validate computational fluid dynamics simulations [10]. Long-term monitoring campaigns have enabled researchers to quantify seasonal variations in wind characteristics [11,14], assess the influence of local topography on wind flow patterns [15], and develop refined wind load specifications for bridge codes [16,17]. These comprehensive analyses have demonstrated that SHM-derived wind data provide invaluable information for understanding the intricate aerodynamic environment of bridge sites, supporting both design verification and operational safety assessment.

While numerous studies have utilized field-measured wind data to characterize wind field parameters, the accuracy and reliability of such analyses fundamentally hinge on the quality of the monitoring data. The measured wind speed and direction are often assumed to accurately represent the undisturbed atmospheric wind conditions to which the bridge structure is exposed [18,19,20]. However, due to the complex aerodynamic interactions between the bridge deck and the surrounding airflow, these measurements can be significantly affected by localized disturbances induced by the deck itself [21,22]. Additionally, chaotic structural responses, which may arise from the inherent flexibility and dynamic characteristics of long-span bridges, might also couple with aerodynamic disturbances, further complicating the disturbance effects on wind measurements [23]. Such disturbances may lead to systematic biases in the recorded wind speed and direction, resulting in anomalous data distributions and potentially inaccurate wind load estimations.

These disturbance effects have been observed on various long-span bridges [13,14,15]. However, these early studies primarily identified the phenomena without providing in-depth analysis or systematic solutions for the measurement bias. The first comprehensive investigation was conducted by Ma et al. [21], who emphasized the necessity of accounting for deck-induced disturbances in wind field characterization and proposed a statistical model to quantify their influence on wind parameters for the Sutong Cable-stayed Bridge. While this pioneering work established the theoretical foundation for disturbance analysis, its applicability remained limited to cable-stayed bridge configurations. Building on this framework, Chen et al. [22] developed a deep learning-based data-cleaning framework using variational autoencoder networks to reconstruct undistorted wind field data from disturbed measurements during typhoon events at the same Sutong Cable-stayed Bridge. Although this approach demonstrated effectiveness for second-level data cleaning, its application was restricted to extreme weather conditions and cable-stayed bridge structures.

Despite these significant advancements, current research efforts have been predominantly confined to cable-stayed bridges, lacking systematic investigation into disturbance effects for different bridge types, particularly long-span suspension bridges. This research gap is particularly significant considering that suspension bridges and cable-stayed bridges exhibit fundamentally different aerodynamic characteristics due to their distinct structural configurations and flow interaction mechanisms [24,25,26]. Furthermore, geographical and environmental contexts vary substantially across different bridge sites, leading to site-specific wind field properties that can significantly influence the magnitude and patterns of deck disturbance effects [14,27]. The extension of disturbance-correction methodologies to suspension bridges and different environmental conditions is essential not only for providing a robust and powerful solution for accurate wind field characterization of a specific bridge type but also for validating the generalizability of this approach across diverse bridge configurations and wind environments.

This study takes the Runyang Suspension Bridge as a case study to investigate the disturbance effects of the bridge deck on field-measured wind data and to develop a data-cleaning methodology to derive the actual wind field parameters. The remainder of this paper is organized as follows: Section 2 presents the site-specific wind data collection details and the derivation process of wind field parameters. Section 3 provides a comparative analysis of the wind field parameters and identifies the disturbance effects of the bridge deck on the recorded wind data. A novel component-decomposition method for data cleaning of wind field parameters is proposed in Section 4 to mitigate the disturbance effects and derive the actual wind field parameters, with detailed results and validation also presented in this section. Section 5 presents a probabilistic analysis of the wind field parameters based on the cleaned data, with results compared to the original data to demonstrate the effectiveness of the proposed method. Finally, Section 6 summarizes this study and provides concluding remarks.

## 2. Engineering Background for Wind Data Collection

The Runyang Suspension Bridge, which connects Zhenjiang and Yangzhou across the Yangtze River, is a critical transportation infrastructure in Jiangsu Province, China. With a main span of 1490 m, the bridge is highly susceptible to wind loads. Located near the Yangtze River estuary and influenced by the humid subtropical monsoon climate, the bridge site experiences complex wind environments characterized by seasonal monsoon patterns and frequent typhoon activities [18]. These challenging meteorological conditions, including strong winds from seasonal circulation and extreme typhoon events, emphasize the critical importance of accurate wind field monitoring for structural safety assessment. To monitor wind field characteristics at the bridge site, the structural health monitoring system incorporates four ultrasonic anemometers, with two positioned on opposite sides of the midspan deck (upstream and downstream) and the remaining two installed atop the bridge towers, as illustrated in Figure 1. The strategic placement of these anemometers follows established practices in bridge wind monitoring, targeting the most critical locations for wind-induced response assessments [6,18]. The tower-top locations typically experience the highest wind speeds due to increased elevation and reduced boundary layer effects, while the midspan deck represents the most vulnerable section for wind-induced vibrations and aerodynamic instability in long-span suspension bridges [27]. Although additional anemometers could be installed at other deck locations, these four positions capture the essential wind characteristics needed for structural safety evaluation and are considered sufficient for comprehensive wind field characterization. This study focuses on the wind field characteristics surrounding the bridge girder. Accordingly, wind data collected by the anemometers on the upstream and downstream sides of the midspan deck (referred to as ANED and ANEU, respectively) are analyzed. Figure 2 illustrates the anemometer layout on the midspan deck. These anemometers record the wind speed ($U_{r}$) and wind direction (β) in the horizontal plane at a sampling frequency of 1 Hz, with the wind direction measured clockwise and $0^{\circ }$ representing true north.

To extract wind field parameters from the measured data, we employ the vector-based approach for analyzing turbulence characteristics [28]. The process begins by transforming the recorded wind speed $U_{r}$ from both ANEU and ANED into orthogonal components ($U_{x}$ and $U_{y}$) through trigonometric decomposition, where(1)$$\begin{array}{c} U_{x}=U_{r}\cos\beta , \end{array}$$(2)$$\begin{array}{c} U_{y}=U_{r}\sin\beta . \end{array}$$
Subsequently, the time-averaged wind speed magnitude and its associated direction ($\bar{U}$ and $\bar{\beta }$) are calculated using vector algebra, where(3)$$\begin{array}{c} \bar{U}=\sqrt{{\bar{U}}_{x}^{2}+{\bar{U}}_{y}^{2}}, \end{array}$$(4)$$\begin{array}{c} \bar{\beta }=\arctan\frac{{\bar{U}}_{y}}{{\bar{U}}_{x}}. \end{array}$$
Following standard practice, a 10 min averaging window is adopted to characterize both mean and turbulent wind properties [29,30]. Within this framework, the turbulent wind velocity fluctuations in the streamwise *u* and cross-stream *v* directions are obtained through coordinate transformations, where(5)$$\begin{array}{c} u=U_{x}\cos\bar{\beta }+U_{y}\sin\bar{\beta }-\bar{U}, \end{array}$$(6)$$\begin{array}{c} v=-U_{x}\sin\bar{\beta }+U_{y}\cos\bar{\beta }. \end{array}$$
Wind flow variability is quantified through turbulence intensity, which is calculated as the normalized standard deviation (SD) of velocity fluctuations relative to the mean wind speed, with(7)$$\begin{array}{c} I_{u}=\frac{\sigma_{u}}{\bar{U}}, \end{array}$$(8)$$\begin{array}{c} I_{v}=\frac{\sigma_{v}}{\bar{U}}, \end{array}$$
where $\sigma_{u}$ and $\sigma_{v}$ denote the standard deviations of the *u* and *v* components, respectively. Throughout this analysis, turbulence intensity refers specifically to the streamwise component, which is designated as *I*.

For the subsequent analysis, we perform a coordinate transformation to align all wind direction measurements with the bridge axis as the reference frame. Given that the bridge orientation deviates $21.5^{\circ }$ west from true north, the transformed wind direction α (°) is computed as(9)$$\alpha^{\prime }=\delta \left(\bar{\beta }-\frac{21.5}{180}\pi \right).$$
To maintain consistency within the standard angular range of [0,360), any transformed direction values exceeding this range are normalized using(10)$$\alpha =\delta \left(\alpha^{\prime }\right)=\alpha^{\prime }-\left\lfloor \frac{\alpha^{\prime }}{2\pi }\right\rfloor \times 2\pi ,$$
where $\left\lfloor \cdot \right\rfloor$ represents the floor function for downward rounding.

## 3. Recognition and Explanation of Bridge Deck Disturbance Effects

To elucidate the disturbance effects of the bridge deck on wind field parameters, we conducted a comparative analysis of the synchronous 10 min mean wind speed and turbulence intensity recorded by ANED and ANEU, as shown in Figure 3a and Figure 3b, respectively. Here, the wind direction classification is based on the bridge-axis coordinate system, where upstream wind corresponds to the angular range of $\left(180^{\circ },360^{\circ }\right)$ (approximately westward directions) and downstream wind corresponds to $\left(0^{\circ },180^{\circ }\right)$ (approximately eastward directions). The comparative plots reveal distinct trends: clusters of data points, which are distinguished by color (blue for upstream wind and red for downstream wind), exhibit systematic deviations from the ideal y=x line. Specifically, for the mean wind speed, red data points are predominantly located below the y=x line, while blue data points are mostly above it. In contrast, for turbulence intensity, the distribution is reversed, with red data points appearing above and blue data points appearing below the y=x line. These findings highlight a notable discrepancy in the wind field parameters between ANED and ANEU. For upstream wind, the mean wind speed measured by ANED is lower than that recorded by ANEU, while the turbulence intensity is higher. Conversely, for downstream wind, the mean wind speed is higher, and the turbulence intensity is lower at ANED compared to ANEU. Additionally, it is observed that the amplitude of the discrepancy on turbulence intensity is more pronounced than that on the mean wind speed. To quantify these discrepancies, we evaluate the relative root mean square error (RMSE) and mean absolute error (MAE) between ANED and ANEU measurements, using their average as the reference. The analysis reveals significant biases: for wind speed, the relative RMSE and MAE are 15.72% and 12.27%, respectively, while for turbulence intensity, these metrics reach 35.96% and 31.08%, confirming that disturbance effects are more pronounced for turbulence intensity measurements.

The systematic variations in mean wind speed and turbulence intensity between ANED and ANEU measurements stem from the aerodynamic interference generated by the bridge deck structure, as depicted in Figure 4. For wind approaching from the downstream direction, ANED initially captures relatively undisturbed flow conditions. Subsequently, as the airstream encounters the bridge deck geometry, complex flow modifications that fundamentally alter both velocity magnitude and directional characteristics occur. These aerodynamic interactions produce a characteristic pattern of velocity reduction coupled with increased turbulence. The modified airflow subsequently propagates upstream where it is captured by ANEU, thereby generating the observed parameter disparities between measurement locations. For upstream-originating winds, an analogous disturbance mechanism manifests in a reverse sequence. When wind directions are approximately aligned with the bridge longitudinal axis, wind flow experiences disturbance effects from both sides of the deck; thus both anemometers (ANED and ANEU) record similarly affected wind measurements. However, since the most critical wind-induced phenomena for long-span bridges are driven by the wind component perpendicular to the bridge deck, the disturbance effects along the bridge axis are of secondary concern for structural safety assessment [1,2,3]. It is worth noting that these aerodynamic disturbances may be further complicated by the nonlinear and potentially chaotic dynamic responses of the flexible bridge structure itself, which could introduce additional sources of variability in the measured wind parameters [23].

It should be noted that the statistical quantities such as the mean wind speed and turbulence intensity calculated over the 10 min averaging window demonstrate systematic biases due to deck-induced disturbances. However, for instantaneous wind speeds and directions, the disturbance patterns may exhibit complex and variable characteristics that do not necessarily show consistent effects at every moment. The temporal averaging process helps capture these systematic statistical trends while filtering out short-term fluctuations. For investigations of disturbance effects at shorter time scales, more sophisticated approaches may be required, as discussed in our related work [22].

Hence, it is evident that the bridge deck disturbance significantly impacts the wind field parameters recorded by field anemometers. To accurately investigate the wind field characteristics at the bridge site, it is essential to develop a data-cleaning method that can effectively mitigate the disturbance effects and derive the actual wind field parameters.

## 4. Data Cleaning for Wind Field Parameters

This section presents a component-decomposition method for the data cleaning of wind field parameters, which aims to isolate the disturbance component and establish statistical models to predict the actual wind field parameters based on the disturbed data. The methodological framework adopted in this study builds upon our previous work on component decomposition for cable-stayed bridges [21] and extends its application to long-span suspension bridges. The following subsections detail the proposed method and its implementation.

### 4.1. Description of the Proposed Method

Based on the findings in Section 3, bridge deck disturbance leads to a reduction in wind speed. The correlation between the 10 min mean wind speed derived from anemometer measurements and the undisturbed 10 min mean wind speed can be formulated as(11)$$\bar{U}={\bar{U}}_{a}-{\bar{U}}_{d},$$
where $\bar{U}$ denotes the 10 min mean wind speed from anemometer measurements, ${\bar{U}}_{a}$ signifies the undisturbed (actual) 10 min mean wind speed, and ${\bar{U}}_{d}$ characterizes the deviation between $\bar{U}$ and ${\bar{U}}_{a}$ resulting from bridge deck disturbance. Given the approximately linear correlation between these deviations and the undisturbed mean wind speed as demonstrated in Figure 3a, we introduce a non-negative coefficient ($f_{u}$) to characterize the magnitude of bridge deck disturbance on the undisturbed mean wind speed. Subsequently, ${\bar{U}}_{d}$ is calculated as(12)$${\bar{U}}_{d}={\bar{U}}_{a}\cdot f_{u,i}\left(\alpha_{n}\right),$$
where $f_{u,i}\left(\alpha_{n}\right)$ denotes the disturbance coefficient for the mean wind speed of the *i*th anemometer, which depends solely on $\alpha_{n}$. The parameter $\alpha_{n}$ represents the nominal relative mean wind direction, which is calculated as the vector average of the relative mean wind direction from ANED and ANEU and expressed as(13)$$\alpha_{n}=\arctan\left[\frac{\sin\alpha_{\mathrm{ANED}}+\sin\alpha_{\mathrm{ANEU}}}{\cos\alpha_{\mathrm{ANED}}+\cos\alpha_{\mathrm{ANEU}}}\right].$$
Given the structural symmetry of the midspan bridge deck, the disturbance coefficients for ANED and ANEU (designated as $f_{u,\mathrm{ANED}}\left(\alpha_{n}\right)$ and $f_{u,\mathrm{ANEU}}\left(\alpha_{n}\right)$, respectively) exhibit symmetrical behavior. Therefore, the correlation between $f_{u,\mathrm{ANED}}$ and $f_{u,\mathrm{ANEU}}$ can be expressed as(14)$$f_{u,\mathrm{ANEU}}\left(\alpha_{n}\right)=f_{u,\mathrm{ANED}}\left[\delta (\alpha_{n}+\pi )\right],$$
where δ follows the definition in Equation (Equation 10). For continuity requirements, $f_{u,\mathrm{ANED}}$ must satisfy(15)$$\lim_{\alpha_{n}\to {2\pi }^{-}}f_{u,\mathrm{ANED}}\left(\alpha_{n}\right)=f_{u,\mathrm{ANED}}\left(0\right)\;\mathrm{For}\;0\leq \alpha_{n}<2\pi .$$
In the present investigation, a high-order polynomial function is employed to approximate $f_{u,\mathrm{ANED}}$. The choice of polynomial formulation is motivated by its universal approximation capability for smooth functions and computational efficiency [31,32]. Given that the symmetry and continuity constraints (Equations (Equation 14) and (Equation 15)) have been explicitly encoded, the remaining functional characteristics are relatively straightforward, making a fourth-order polynomial sufficient for accurate representation. The polynomial is formulated as(16)$${\hat{f}}_{u,\mathrm{ANED}}\left(\alpha_{n}\right)=\left|\alpha_{n}\left(\alpha_{n}-2\pi \right)\left(p_{u1}\alpha_{n}^{2}+p_{u2}\alpha_{n}+p_{u3}\right)+p_{u4}\right|\;\mathrm{For}\;0\leq \alpha_{n}<2\pi ,$$
where ${\hat{f}}_{u,\mathrm{ANED}}\left(\alpha_{n}\right)$ denotes the estimated value of $f_{u,\mathrm{ANED}}$, the caret notation indicates predicted values, and $p_{ui}$(i=1,2,3,4) represents calibration parameters for wind speed modeling. Consequently, the predicted 10 min mean wind speed for ANED and ANEU can be expressed, respectively, as(17)$$\begin{array}{c} {\hat{\bar{U}}}_{\mathrm{ANED}}={\bar{U}}_{a}\cdot \left[1-{\hat{f}}_{u,\mathrm{ANED}}\left(\alpha_{n}\right)\right], \end{array}$$(18)$$\begin{array}{c} {\hat{\bar{U}}}_{\mathrm{ANEU}}={\bar{U}}_{a}\cdot \left\{1-{\hat{f}}_{u,\mathrm{ANED}}\left[\delta (\alpha_{n}+\pi )\right]\right\}. \end{array}$$
The calibration parameters can be determined through a standard nonlinear optimization framework that minimizes the sum of squared errors between model predictions and field observations and are formulated as(19)$$\begin{array}{cc} \mathrm{Objective}\;\mathrm{function}: & \;{\hat{p}}_{ui}=\underset{p_{ui}}{\arg\min}\sum_{i=1}^{N}{\left|y_{i}-{\hat{y}}_{i}\right|}^{2}\;, \end{array}$$(20)$$\begin{array}{cc} \mathrm{Model}\;\mathrm{prediction}: & \;{\hat{y}}_{i}=\frac{{\hat{\bar{U}}}_{\mathrm{ANED}}}{{\hat{\bar{U}}}_{\mathrm{ANEU}}}=\frac{1-{\hat{f}}_{u,\mathrm{ANED}}\left(\alpha_{n,t_{i}}\right)}{1-{\hat{f}}_{u,\mathrm{ANED}}\left[\delta (\alpha_{n,t_{i}}+\pi )\right]}, \end{array}$$(21)$$\begin{array}{cc} \mathrm{Filed}\;\mathrm{measurement}: & \;y_{i}=\frac{{\bar{U}}_{\mathrm{ANED}}\left(t_{i}\right)}{{\bar{U}}_{\mathrm{ANEU}}\left(t_{i}\right)}, \end{array}$$
where i=1,2,…,N and *N* represents the total number of observations. Through identification of model parameters $p_{ui}$ from Equations (Equation 19)–(21), the disturbance coefficient for ANED and ANEU can be predicted using Equations (Equation 14) and (Equation 16). Subsequently, the undisturbed 10 min mean wind speed for the *i*th anemometer (ANED or ANEU) can be estimated as(22)$${\hat{\bar{U}}}_{a,i}=\frac{{\bar{U}}_{i}}{1-{\hat{f}}_{u,i}\left(\alpha_{n}\right)}.$$
Utilizing Equation (Equation 22), the 10 min undisturbed mean wind speed for ANED and ANEU (i.e., ${\hat{\bar{U}}}_{a,\mathrm{ANED}}$ and ${\hat{\bar{U}}}_{a,\mathrm{ANEU}}$) can be determined. Finally, the average of ${\hat{\bar{U}}}_{a,\mathrm{ANED}}$ and ${\hat{\bar{U}}}_{a,\mathrm{ANEU}}$ is adopted as the representative 10 min undisturbed mean wind speed, which is defined as(23)$${\hat{\bar{U}}}_{a}=\frac{{\hat{\bar{U}}}_{a,\mathrm{ANED}}+{\hat{\bar{U}}}_{a,\mathrm{ANEU}}}{2}=\frac{1}{2}\left(\frac{{\bar{U}}_{\mathrm{ANED}}}{1-{\hat{f}}_{u,\mathrm{ANED}}\left(\alpha_{n}\right)}+\frac{{\bar{U}}_{\mathrm{ANEU}}}{1-{\hat{f}}_{u,\mathrm{ANEU}}\left(\alpha_{n}\right)}\right).$$

Bridge deck disturbance results in an elevation in turbulence intensity. Consequently, the disturbed component decomposition formulation for turbulence intensity *I* can be represented as(24)$$I=I_{a}+I_{d}=I_{a}\left(1+f_{I,i}\left(\alpha_{n}\right)\right),$$
where $I_{a}$ denotes the undisturbed turbulence intensity, $I_{d}$ represents the disturbance component, and $f_{I,i}\left(\alpha_{n}\right)$ characterizes the disturbance coefficient associated with turbulence intensity for the *i*th anemometer.

The formulation of the trial function for the disturbance coefficient at the downstream position $f_{I,\mathrm{ANED}}$ and upstream position $f_{I,\mathrm{ANEU}}$, along with the construction of optimization problems, follows a similar approach to that employed for the mean wind speed. Therefore, the nonlinear optimization problem can be expressed as(25)$$\begin{array}{cc} \mathrm{Objective}\;\mathrm{function}: & \;{\hat{p}}_{Ii}=\underset{p_{Ii}}{\arg\min}\sum_{i=1}^{N}{\left|y_{i}-{\hat{y}}_{i}\right|}^{2}\;, \end{array}$$(26)$$\begin{array}{cc} \mathrm{Model}\;\mathrm{prediction}: & \;{\hat{y}}_{i}=\frac{{\hat{I}}_{\mathrm{ANED}}}{{\hat{I}}_{\mathrm{ANEU}}}=\frac{1+{\hat{f}}_{I,\mathrm{ANED}}\left(\alpha_{n,t_{i}}\right)}{1+{\hat{f}}_{I,\mathrm{ANED}}\left[\delta (\alpha_{n,t_{i}}+\pi )\right]}, \end{array}$$(27)$$\begin{array}{cc} \mathrm{Filed}\;\mathrm{measurement}: & \;y_{i}=\frac{I_{\mathrm{ANED}}\left(t_{i}\right)}{I_{\mathrm{ANEU}}\left(t_{i}\right)}, \end{array}$$
where i=1,2,…,N and *N* denotes the total number of observations. The representative undisturbed turbulence intensity can be estimated as(28)$${\hat{I}}_{a}=\frac{{\hat{I}}_{a,\mathrm{ANED}}+{\hat{I}}_{a,\mathrm{ANEU}}}{2}=\frac{1}{2}\left(\frac{I_{\mathrm{ANED}}}{1+{\hat{f}}_{I,\mathrm{ANED}}\left(\alpha_{n}\right)}+\frac{I_{\mathrm{ANEU}}}{1+{\hat{f}}_{I,\mathrm{ANEU}}\left(\alpha_{n}\right)}\right).$$

### 4.2. Model Validation and Performance Evaluation

To validate the proposed data-cleaning method, it is applied to wind field monitoring data collected from the Runyang Suspension Bridge in 2024. As detailed in Section 2, these data are obtained from the structural health monitoring system equipped with four ultrasonic anemometers, with a specific focus on measurements from the upstream and downstream anemometers (ANED and ANEU) positioned at the midspan deck. To avoid numerical instability caused by low wind speeds, only data with mean wind speeds greater than 2 m/s are considered for analysis.

For the mean wind speed, the model parameters ($p_{ui}$) are estimated as(29)$$p_{u1}=0.0467,\;p_{u2}=-0.4314,\;p_{u3}=1.0058,\;p_{u4}=-1.2558.$$
The observations and predictions of the optimization problem are shown in Figure 5a, suggesting a general agreement between the model predictions and observations. The fitted curve of ${\hat{f}}_{u,\mathrm{ANED}}$ and ${\hat{f}}_{u,\mathrm{ANEU}}$ is presented in Figure 5b. It is revealed that, even in the absence of explicit constraints, the fitted disturbance coefficient of ANED attains its minimum value precisely at the nominal relative mean wind direction $\alpha_{n}=90^{\circ }$. For ANEU, the opposite is observed. These findings are consistent with the analysis of bridge deck disturbance theory presented in Section 3.

The coefficient of determination $R^{2}$ [33] is adopted as the measure of fit quality, which is defined as(30)$$R^{2}=1-\frac{\sum_{i=1}^{N}{\left(y_{i}-{\hat{y}}_{i}\right)}^{2}}{\sum_{i=1}^{N}{\left(y_{i}-\bar{y}\right)}^{2}},$$
where $\bar{y}$ is the mean value of the observations ($y_{i}$). The $R^{2}$ value for the mean wind speed is 0.778, indicating a good fit between the model predictions and observations.

The actual mean wind speed for ANED and ANEU can then be reconstructed using the derived disturbance coefficients, as illustrated in Figure 6a and Figure 6b, respectively. In Figure 6a, it is found that the corrected measurements from the downstream side closely match the original observed data. In contrast, the corrected readings from the upstream side, which reflect the primarily disturbed wind conditions, show higher values than the observed data. Similar trends are evident for the corrected mean wind speed of ANEU, as shown in Figure 6b. These results confirm the effectiveness of the proposed data-cleaning approach in reducing disturbance effects and obtaining the actual wind field parameters.

Similarly, the model parameters ($p_{Ii}$) for turbulence intensity are estimated as(31)$$p_{I1}=-0.0013,\;p_{I2}=0.0875,\;p_{I3}=-0.2951,\;p_{I4}=-1.1650.$$
with an $R^{2}$ value of 0.7042. The model prediction and observations are shown in Figure 7a, and the fitted curve of ${\hat{f}}_{I,\mathrm{ANED}}$ and ${\hat{f}}_{I,\mathrm{ANEU}}$ is presented in Figure 7b. Similar to the disturbance coefficient of the mean wind speed, the fitted disturbance coefficient of turbulence intensity for ANED attains its minimum value at $\alpha_{n}=90^{\circ }$ and approaches its maximum value at $\alpha_{n}=270^{\circ }$, while the opposite is observed for ANEU.

To verify the robustness of our optimization approach, we conducted a comparative analysis of different optimization algorithms for turbulence intensity disturbance coefficient identification. Four widely used optimization algorithms were evaluated: the Levenberg–Marquardt [34], Trust Region Reflective [35], Nelder–Mead [36], and Powell [37] methods. As shown in Figure 8, three algorithms (Levenberg–Marquardt, Trust Region Reflective, and Powell) converged to essentially identical solutions, with the only exception being the Nelder–Mead algorithm, which exhibited slight deviations at peak positions. This results demonstrate the overall robustness of the optimization process and the reliability of the identified disturbance coefficients.

The corrected turbulence intensity of ANED and ANEU is shown in Figure 9a and Figure 9b, respectively. In Figure 9a, the corrected turbulence intensity of the downstream side closely aligns with the initially observed data, while the corrected turbulence intensity of the upstream side is significantly lower than the observed data, where the disturbance effect is more pronounced. A similar observation can be made for the corrected turbulence intensity of ANEU, as shown in Figure 9b.

To evaluate the performance of the proposed data-cleaning method, we analyze the relative error between ANED and ANEU measurements across different wind direction sectors, as shown in Figure 10. For the mean wind speed (which exhibits a decrease under disturbance), the maximum value serves as the reference; for turbulence intensity (which exhibits an increase under disturbance), the minimum value serves as the reference. Since both anemometers experience deck-induced disturbances when the wind direction aligns with the bridge axis, validation focuses on sectors $\left[45^{\circ },135^{\circ }\right]$ and $\left[225^{\circ },315^{\circ }\right]$. Within these sectors, the relative error for the mean wind speed is controlled within 5%, while the turbulence intensity error remains below 10%, demonstrating satisfactory performance of the proposed method. Notably, the error distribution across different wind directions exhibits a systematic pattern that correlates with the angular position, which is consistent with the symmetry constraint imposed by Equation (Equation 14), confirming the validity of the theoretical framework underlying the component-decomposition approach.

We present the comparative diagram of corrected synchronous wind field parameters between ANED and ANEU in Figure 11a,b. It can be observed that, for both mean wind speed and turbulence intensity, compared to the original data shown in Figure 3a,b, the corrected data of ANED and ANEU exhibit a high degree of consistency with no noticeable discrepancies, which provides strong evidence of the validity of the proposed data-cleaning method.

## 5. Probabilistic Analysis of Corrected Wind Field Parameters

Based on the corrected wind field parameters, we conduct a probabilistic analysis of the wind speed and turbulence intensity. For the mean wind speed, we fit the corrected data to a Weibull distribution, which has been widely validated for modeling wind speed data due to its flexibility in capturing the characteristic shape of wind speed distributions across different meteorological conditions [38,39]. The probability density function (PDF) of the Weibull distribution is given by(32)$$f\left(s;\xi \right)=\xi s^{\xi -1}\exp\left(-s^{\xi }\right)$$
where $s=\frac{x-\mu }{\sigma }$, μ is the location parameter, σ is the scale parameter, and ξ is the shape parameter.

For turbulence intensity, we fit the corrected data to a Generalized Extreme Value (GEV) distribution. The GEV distribution is particularly suitable for modeling turbulence intensity as it can effectively capture the asymmetric and bounded nature of turbulence data, which often exhibits extreme value characteristics in atmospheric boundary-layer flows [40,41]. The PDF of the GEV distribution is given by(33)$$f\left(s;\xi \right)=\left\{\begin{array}{cc} \exp\left(-s\right)\exp\left[-\exp\left(-s\right)\right] & \mathrm{if}\;\xi =0,\\ {\left(1-\xi s\right)}^{1/\xi -1}\exp\left[-{\left(1-\xi s\right)}^{1/\xi }\right] & \mathrm{if}\;\xi \neq 0\;\mathrm{and}\;\xi s<1,\\ 0 & \mathrm{otherwise}, \end{array}\right.$$
where $s=\frac{x-\mu }{\sigma }$, μ is the location parameter, σ is the scale parameter, and ξ is the shape parameter. The parameters of the Weibull and GEV distributions are estimated using the maximum likelihood estimation (MLE) [42] method.

Figure 12 and Figure 13 present the comparative probability density functions (PDFs) of the uncorrected and corrected mean wind speed and turbulence intensity, respectively. For turbulence intensity, the corrected data exhibits a more concentrated distribution, and the peak value of the PDF curve is lower than that of the uncorrected data, indicating a more stable wind flow. For mean wind speed, although the concentration degree dose not change significantly, the PDF curve of the corrected data is slightly shifted to higher values compared to the uncorrected data. To assess the goodness of fit of the selected distributions, we evaluate the PDF R-squared ($R_{\mathrm{PDF}}^{2}$) and quantile R-squared ($R_{\mathrm{quantile}}^{2}$) metrics [43,44]. For mean wind speed, the Weibull distribution achieves $R_{\mathrm{PDF}}^{2}=0.9872$ and $R_{\mathrm{quantile}}^{2}=0.9996$, while for turbulence intensity, the GEV distribution yields $R_{\mathrm{PDF}}^{2}=0.9773$ and $R_{\mathrm{quantile}}^{2}=0.9654$, indicating excellent agreement between the fitted distributions and the data.

Table 1 presents the mean wind speed of different return periods, which are estimated based on the corrected mean wind speed using the fitted Weibull distribution model. Compared to the uncorrected mean wind speed of ANED and ANEU, the corrected mean wind speed is higher for all return periods, with a typical value of 19.197 m/s for a 100-year return period. This indicates that the disturbance effects of the bridge deck lead to an underestimation of the mean wind speed, which is crucial for the design and assessment of wind loads on the structure. The higher corrected wind speeds have significant implications for bridge design loads and code compliance, as they suggest that existing structures may experience greater wind loads than previously estimated, potentially necessitating design reassessment or strengthening measures.

The joint distribution characteristics between the corrected wind speed and turbulence intensity are further investigated. Figure 14c displays the scatter plot of these corrected parameters. To facilitate the comparison, scatter plots for the uncorrected data from both ANED and ANEU are also provided in Figure 14a and Figure 14b, respectively. A clear stratification pattern can be observed in the uncorrected turbulence intensity data with respect to mean wind speed, where ANED and ANEU exhibit opposite stratification orders. However, such directional dependence disappears in the corrected data distribution. The turbulence intensity values show wide dispersion at lower wind speeds, but this scatter progressively diminishes as wind speed increases, with values eventually concentrating within a narrow band near 0.08. The effectiveness of the proposed data-cleaning methodology is further substantiated by these findings.

## 6. Conclusions Remarks

This study presents a comparative analysis of wind field parameters recorded by anemometers positioned on the upstream and downstream sides of the midspan deck of the Runyang Suspension Bridge, revealing the disturbance effects of the bridge deck on the wind data. To mitigate these disturbance effects and derive the actual wind field parameters, we propose a novel data-cleaning method that decomposes the disturbed components from the recorded data and establishes a statistical model to bridge the gap between the disturbed and actual wind field parameters. The results demonstrate that the proposed method effectively mitigates the disturbance effects caused by the bridge deck, enabling more accurate evaluation of wind loads acting on the structure.

Several limitations of this study should be acknowledged:The investigation is based on data from a single bridge structure (Runyang Suspension Bridge), limiting direct generalizability to other suspension bridges. However, the methodological framework is inherently general and adaptable. Our previous research [21] has successfully applied similar approaches to cable-stayed bridges, demonstrating broader applicability through site-specific parameter calibration.This study focuses exclusively on the critical midspan section rather than multiple deck locations along the bridge span. Nevertheless, the mathematical formulation and the optimization framework remain consistent, making the approach readily transferable to multiple locations through systematic parameter recalibration.The proposed method relies on empirical field measurements without computational fluid dynamics (CFD) validation. While the statistical approach provides robust practical results, CFD simulations could offer additional insights into the underlying aerodynamic mechanisms. Future work should explore CFD-based validation to enhance the physical understanding of the disturbance effects and further refine the data-cleaning methodology.The current analysis focuses primarily on aerodynamic disturbance effects without considering the potential influence of nonlinear and chaotic structural responses. These chaotic effects could manifest as additional sources of uncertainty in structural health monitoring systems and may interact with the aerodynamic phenomena investigated. Future work could benefit from incorporating considerations of such nonlinear dynamic behavior to further enhance the robustness of wind field characterization methods [23].

Despite these limitations, the methodology provides a systematic framework that can be efficiently adapted to other long-span bridges and multiple measurement locations. Future research should expand validation to multiple bridge sites, implement measurement networks across various deck locations, and explore CFD-based validation of the disturbance mechanisms.

## Figures and Tables

**Figure 1 sensors-25-06503-f001:**
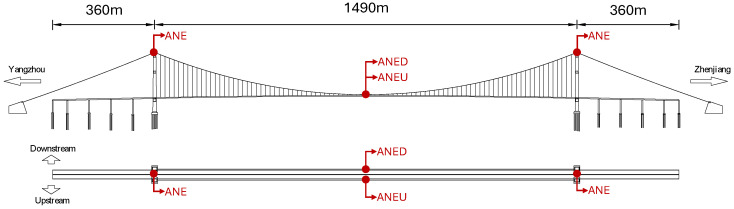
Anemometer layout for wind data collection on the Runyang Suspension Bridge.

**Figure 2 sensors-25-06503-f002:**
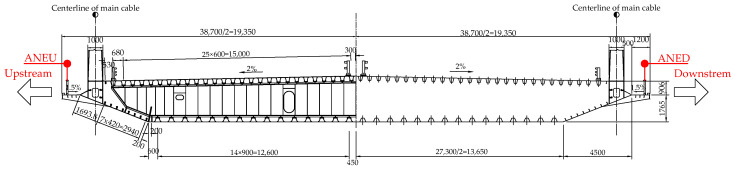
Midspan section of the Runyang Suspension Bridge (mm).

**Figure 3 sensors-25-06503-f003:**
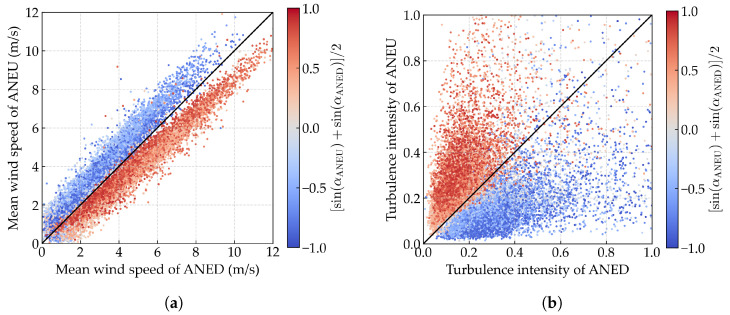
Comparative diagram of synchronous 10 min wind field parameters between ANED and ANEU. The color of the points represents the averaged sine value of the mean wind direction of ANED and ANEU. The blue color indicates a negative sine value for upstream wind, while the red color indicates a positive sine value for downstream wind. The black solid line represents the ideal y=x line. (**a**) Mean wind speed; (**b**) turbulence intensity.

**Figure 4 sensors-25-06503-f004:**
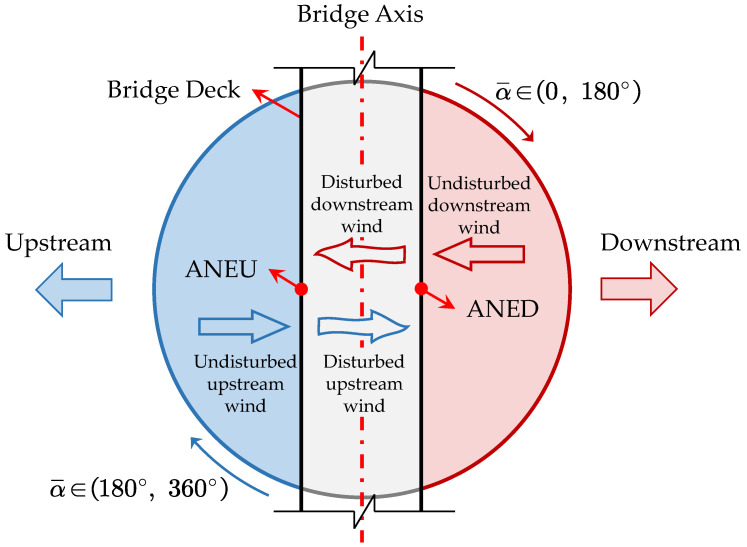
Schematic diagram of bridge deck disturbance effects.

**Figure 5 sensors-25-06503-f005:**
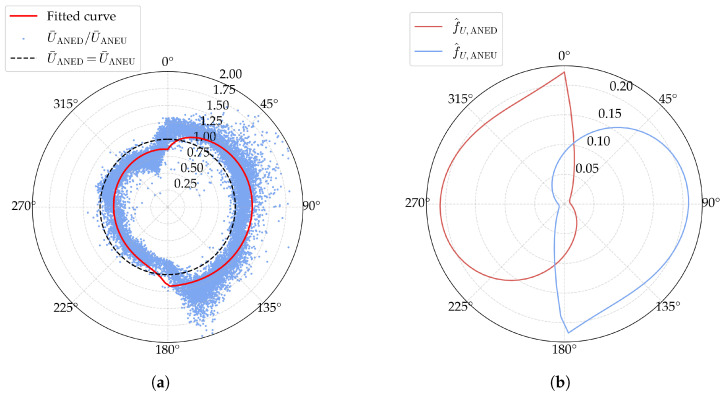
Fitted results of the disturbance coefficient of the mean wind speed: (**a**) observations and predictions; (**b**) the fitted curve of ${\hat{f}}_{u,\mathrm{ANED}}$ and ${\hat{f}}_{u,\mathrm{ANEU}}$.

**Figure 6 sensors-25-06503-f006:**
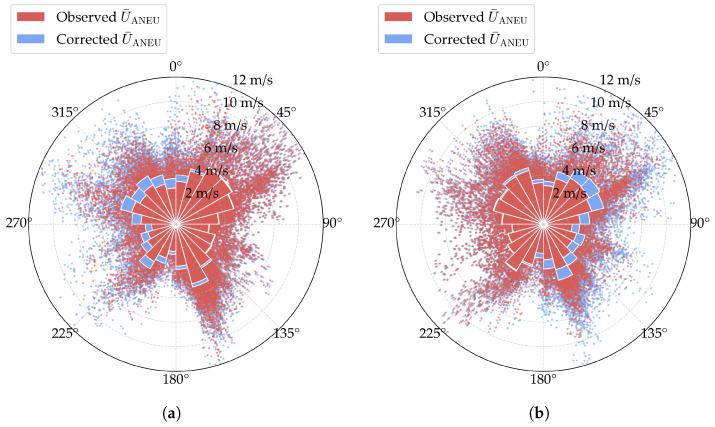
Wind rose diagrams of the corrected mean wind speed. The scatters represent specific samples of the observed or corrected values, while the bars represent the mean values in each wind direction sector. (**a**) ANED; (**b**) ANEU.

**Figure 7 sensors-25-06503-f007:**
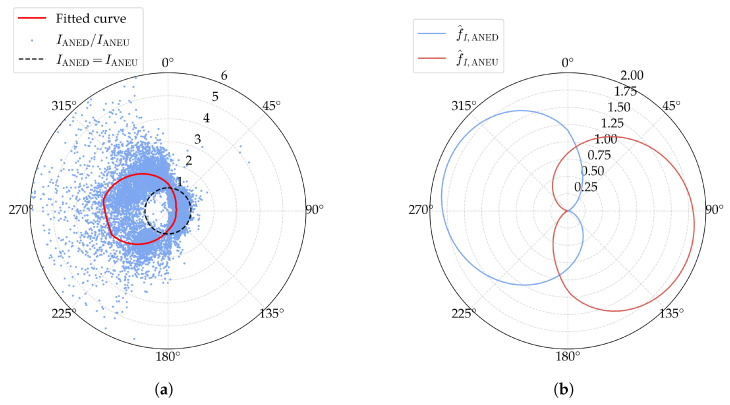
Fitted results of the disturbance coefficient of turbulence intensity: (**a**) observations and predictions; (**b**) the fitted curve of ${\hat{f}}_{I,\mathrm{ANED}}$ and ${\hat{f}}_{I,\mathrm{ANEU}}$.

**Figure 8 sensors-25-06503-f008:**
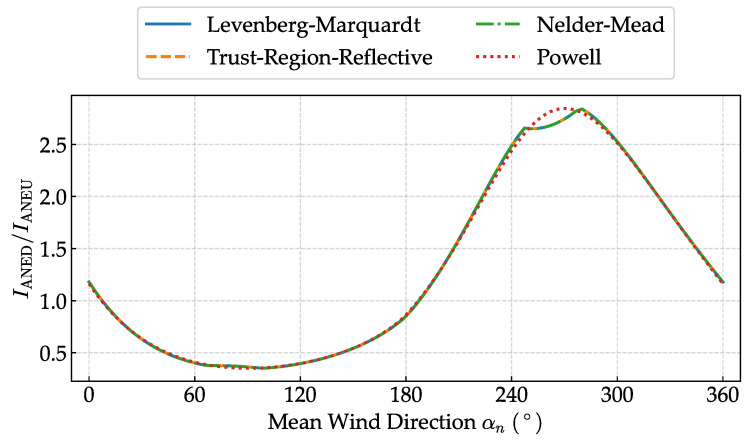
Comparison of the fitted curves for different optimization algorithms.

**Figure 9 sensors-25-06503-f009:**
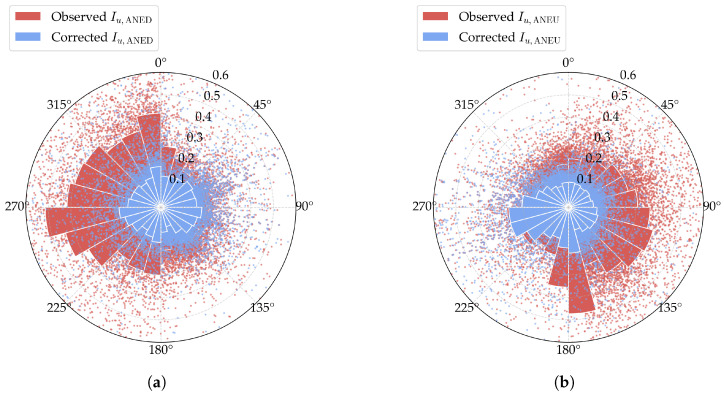
Wind rose diagrams of the corrected turbulence intensity. The scatters represent specific samples of the observed or corrected values, while the bars represent the mean values in each wind direction sector. (**a**) ANED; (**b**) ANEU.

**Figure 10 sensors-25-06503-f010:**
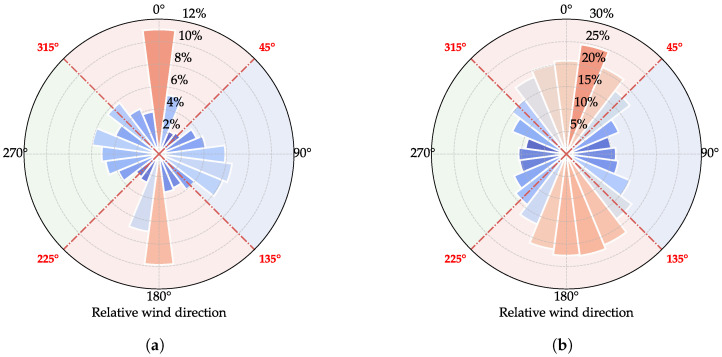
Relative error of wind field parameters between ANED and ANEU within different wind direction sectors. (**a**) Mean wind speed: $\frac{|{\bar{U}}_{\mathrm{ANED}}-{\bar{U}}_{\mathrm{ANEU}}|}{\max\{{\bar{U}}_{\mathrm{ANED}},\;{\bar{U}}_{\mathrm{ANEU}}\}}$. (**b**) Turbulence intensity: $\frac{|I_{\mathrm{ANED}}-I_{\mathrm{ANEU}}|}{\min\{I_{\mathrm{ANED}},\;I_{\mathrm{ANEU}}\}}$. The color of the sectors represents whether the anemometers are disturbed within that wind direction range. Red color indicates that both anemometers are disturbed, while blue and green colors indicate that only one anemometer is disturbed.

**Figure 11 sensors-25-06503-f011:**
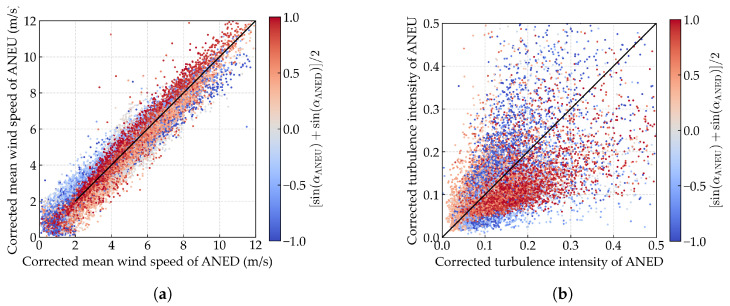
Comparative diagram of corrected synchronous wind field parameters between ANED and ANEU. (**a**) Mean wind speed; (**b**) turbulence intensity.

**Figure 12 sensors-25-06503-f012:**
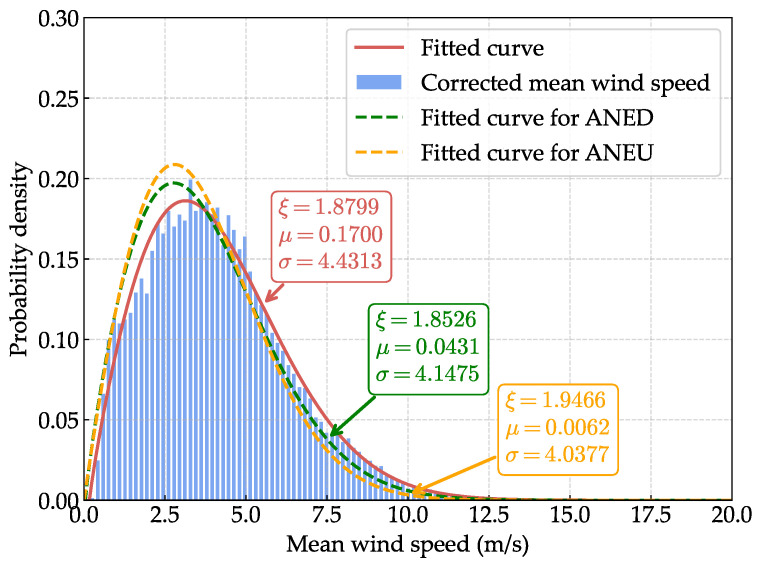
Comparative probability density function (PDF) of the uncorrected and corrected mean wind speeds.

**Figure 13 sensors-25-06503-f013:**
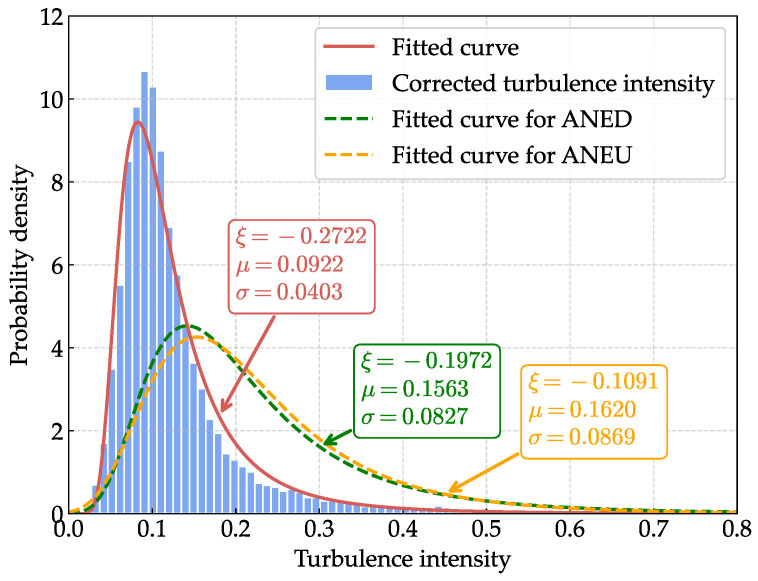
Comparative probability density function (PDF) of the uncorrected and corrected turbulence intensity.

**Figure 14 sensors-25-06503-f014:**
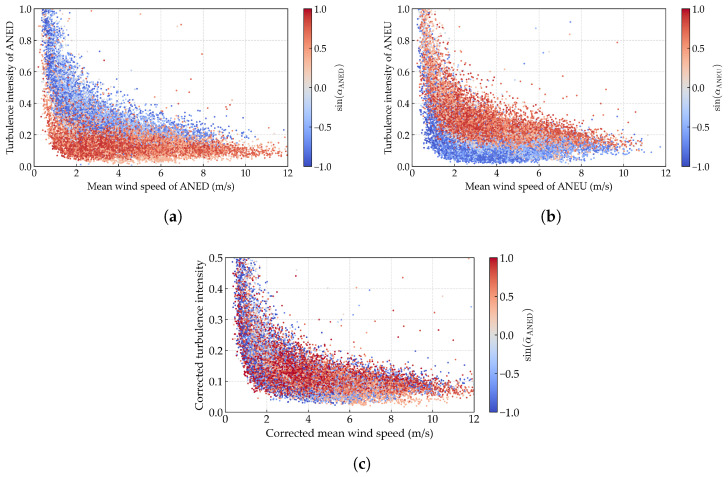
Scatter plot of the mean wind speed versus turbulence intensity: (**a**) uncorrected data of ANED, (**b**) uncorrected data of ANEU, and (**c**) corrected data.

**Table 1 sensors-25-06503-t001:** Wind speed of different return periods (m/s).

Return Period (Year)	5	10	20	50	100
Uncorrected mean wind speed of ANED	16.242	16.722	17.190	17.793	18.237
Uncorrected mean wind speed of ANEU	14.773	15.189	15.594	16.115	16.499
Corrected mean wind speed	17.139	17.634	18.117	18.739	19.197

## Data Availability

Data will be made available upon request.

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
