# Peer review of "Investigation of Wind Field Parameters for Long-Span Suspension Bridge Considering Deck Disturbance Effect"

_sensors, 2025, doi:10.3390/s25216503_

Round 1

Reviewer 1 Report

Comments and Suggestions for Authors

The paper addresses an important problem of deck-induced disturbance in wind field measurements on long-span suspension bridges and proposes a statistical data-cleaning method. The work is technically sound and well-structured, however I have the following points several  require clarification and improvement:

  • The dual anemometers at midspan are limited to one deck section. Discuss the extension of measurements to multiple spans for broader validity.

  • The 10-minute averaging and coordinate transformation assume stationarity. Justify this assumption or discuss its limitations.

  • The disturbance identification is clear, but mainly qualitative. Provide quantitative measures of bias (e.g., mean error or % difference).

  • The polynomial form of the disturbance model appears arbitrary. Justify this choice or consider sensitivity analysis with alternative functions.

  • The validation relies mainly on R² values. Include additional error metrics (e.g., RMSE, MAE) and consider cross-validation with independent data.

  • The probabilistic analysis uses Weibull and GEV distributions only. Please justify exclusion of other candidates and add statistical goodness-of-fit tests.

  • Figures 5–9 are informative but visually dense. It would be better if you could simplify legends and axes to improve clarity.

  • The corrected 100-year wind speed is higher than uncorrected. Discuss implications for bridge design loads and code compliance.

  • The study is based on one bridge (Runyang). Acknowledge limitations and discuss applicability to other suspension bridges.

Reviewer 2 Report

Comments and Suggestions for Authors

This manuscript systematically studies the wind field characteristics of long-span suspension bridges and innovatively proposes a data cleaning method based on component decomposition, effectively solving the measurement deviation problem caused by bridge deck turbulence. The study takes the Runyang Suspension Bridge as a case study and combines on-site measurements with statistical modeling to provide reliable theoretical basis and technical solutions for wind resistant design of the bridge. The paper has a complete structure, scientific methods, and significant theoretical value and engineering significance in its conclusions. It is recommended to further improve the following aspects before considering publication.

  1. Propose disturbance coefficient models (Equations 16 and 24), which separate the bridge deck disturbance effect through component decomposition techniques and solve the limitation of traditional methods ignoring local disturbances. Further explanation is needed on the physical significance of high-order polynomial fitting in equation (16), as well as the sensitivity analysis of parameters (such as pu1-pu4).
  2. It is strongly recommended to compare the effects of different optimization algorithms (such as genetic algorithms) on model parameters (Equations 29, 31) to verify robustness.
  3. The current verification is only based on the 2024 data of Runyang Bridge. It is recommended to supplement other bridges or cross year data to verify universality.
  4. Comparison between CFD simulation results and measured calibration data can be added to enhance the credibility of the method.
  5. Figure 12 shows that the corrected turbulence intensity converges to 0.08, and the correlation between this phenomenon and the aerodynamic characteristics of the bridge needs to be explored.
  6. Suggest analyzing the differences in correction effects under different wind directions (such as whether the symmetry between α n=90 ° and 270 ° is fully established).
  7. A more detailed discussion is needed to compare the advantages and disadvantages with existing research (such as Chen et al. [19]'s deep learning framework), highlighting the irreplaceability of this study.
  8. Some formula numbers are duplicated (such as formula 10 and subsequent formulas) and need to be adjusted uniformly.
  9. There is a spelling error in the "Keywords" field in the abstract ("Data cleaning" should be changed to "Data cleaning").
  10. The research results of the paper have important guiding significance for wind field monitoring of large bridges. It is recommended that the author revise and resubmit the paper to address the above issues. Especially, it is necessary to strengthen the validation of method universality and literature comparison to further enhance academic influence.

Reviewer 3 Report

Comments and Suggestions for Authors

In the reviewed manuscript, the authors are considering a data curing technique which is applicable for the estimation of parameters of the wind field acting on a long-span suspension bridge. Its performance is illustrated based on the data avaliable for the Runyang Suspension Bridge. While the paper content is consistent with the scope of the Sensors journal substantial improvements should be introduced into it.

The main flaw is that the method which is claimed to be proposed by the authors is strongly based on the approach described in Ref. 18. However, this manuscript is mentioned only in the Introduction section, and no proper citations of it are provided by the authors when describing their approach, i.e., section 4 of the current paper almost completely repeats section 5 from Ref. 18.

Some other remarks and questions (as they appear throughout the text):

1) In the introduction section, it might be suggest to add an additional paragraph where recent findings and methods for data curing in the sense of wind parameter evaluation are discussed.

2) It might be plausible if authors provide additional details about typical wind loads (i.e., speed, directions) which are typical for the location of the considered Runyang Suspension Bridge.

3) Line 67: Why such amount of anemometers is considered? Is it possible to install more anemometers on the bridge? How would it affect the obtained results?

4) Could the authors specify what was the direction of the incoming wind flow for the results shown in Fig. 3? Without such information, the corresponding analysis (i.e., lines 105-114) becomes unclear.

5) Line 126: Could the authors somehow illustrate how such influence would look like?

6) Line 143: Will such correlation take place for any time span which is outside this 10-minute time limit considered to obtain the results for Fig. 3?

7) Why 10-minute period is chosen as a reference for all the analysis?

8) Subsection 4.2: Could the authors provide more details about how this monitoring data was collected?

Round 2

Reviewer 1 Report

Comments and Suggestions for Authors

Authors have addressed all comments.

Author Response

Conments1: Authors have addressed all comments.

Response1: We thank reviewer's positive comment on our manuscript.

Reviewer 2 Report

Comments and Suggestions for Authors

The author has responded to or revised all the comments I made during the review in the revised manuscript.

Comments on the Quality of English Language

The author has responded to or revised all the comments I made during the review in the revised manuscript.

Author Response

Comment 1: The author has responded to or revised all the comments I made during the review in the revised manuscript.

Response 1: We thank the reviewer for the positive assessment of our work.

Comment 2: The English could be improved to more clearly express the research.

Response 2: Following the reviewer's feedback, we have thoroughly revised the manuscript to enhance language quality, correct grammatical errors, and improve sentence structure for better readability. All language improvements are marked in the revised manuscript.

Reviewer 3 Report

Comments and Suggestions for Authors

Although the authors have responded to the most of the reviewer's comments, the main one which is about the general novelty of the current manuscript is not responded in a proper way.

Since the methodology for the investigation of wind field parameters is totally the same as in the Ref. 21 there is no sense to repeat it once again in the current paper. To the reviewer's opinion, there are no clear signs of any modification and/or advancement of this technique in the presented paper. Therefore, the authors are strongly suggested either to clearly underline such novelty (except the fact that another bridge type is considered) or completely omit this part and submit their paper as a technical note rather than as a research manuscript.
